# The splicing regulator PTBP1 controls the activity of the transcription factor Pbx1 during neuronal differentiation

Anthony J Linares[1], Chia-Ho Lin[2], Andrey Damianov[2], Katrina L Adams[1,3], Bennett G Novitch[3,4], Douglas L Black[2,4]*

[1]Molecular Biology Institute Graduate Program, University of California, Los Angeles, Los Angeles, United States; [2]Department of Microbiology, Immunology and Molecular Genetics, University of California, Los Angeles, Los Angeles, United States; [3]Department of Neurobiology, David Geffen School of Medicine, University of California, Los Angeles, Los Angeles, United States; [4]Eli and Edythe Broad Center of Regenerative Medicine and Stem Cell Research, David Geffen School of Medicine, University of California, Los Angeles, Los Angeles, United States

**Abstract** The RNA-binding proteins PTBP1 and PTBP2 control programs of alternative splicing during neuronal development. PTBP2 was found to maintain embryonic splicing patterns of many synaptic and cytoskeletal proteins during differentiation of neuronal progenitor cells (NPCs) into early neurons. However, the role of the earlier PTBP1 program in embryonic stem cells (ESCs) and NPCs was not clear. We show that PTBP1 controls a program of neuronal gene expression that includes the transcription factor Pbx1. We identify exons specifically regulated by PTBP1 and not PTBP2 as mouse ESCs differentiate into NPCs. We find that PTBP1 represses Pbx1 exon 7 and the expression of the neuronal Pbx1a isoform in ESCs. Using CRISPR-Cas9 to delete regulatory elements for exon 7, we induce Pbx1a expression in ESCs, finding that this activates transcription of neuronal genes. Thus, PTBP1 controls the activity of Pbx1 to suppress its neuronal transcriptional program prior to induction of NPC development.

*For correspondence: dougb@
microbio.ucla.edu

Competing interest: See
page 20

Reviewing editor: Benjamin J
Blencowe, University of Toronto,
Canada

## Introduction

Alternative splicing is an important form of gene regulation during tissue development. In the mammalian nervous system, large scale changes in splice site choice produce many new mRNAs encoding protein isoforms with different structures and functions that are specific to neurons (*Li et al., 2007*; *Licatalosi and Darnell, 2006*; *Norris and Calarco, 2012a*; *Raj and Blencowe, 2015a*; *Yap and Makeyev, 2013*; *Zheng and Black, 2013*). These splicing patterns are regulated in a temporal and cell-specific manner by the expression of specialized pre-mRNA binding proteins (RBPs) (*Black, 2003*; *Braunschweig et al., 2013*; *Fu and Ares, 2014*; *Lee and Rio, 2015*). Several RBPs have been shown to control the expression of isoforms essential for neuronal development and maturation (*Charizanis et al., 2012*; *Gehman et al., 2011*; *Gehman et al., 2012*; *Li et al., 2014*; *Licatalosi et al., 2012*; *Yano et al., 2010*; *Jensen et al., 2000a*; *Iijima et al., 2011a*; *Ince-Dunn et al., 2012b*; *Quesnel-Vallières et al., 2015b*). However, the complex programs of splicing affecting neuronal development are only beginning to be characterized, and the cellular functions of the regulated isoforms are often poorly understood.

The polypyrimidine tract binding (PTB) proteins, PTBP1 and PTBP2, regulate a large set of splicing events during neuronal differentiation (*Keppetipola et al., 2012*). PTBP1 is expressed in neuronal progenitor cells and many non-neuronal cells. Upon neuronal differentiation, PTBP1 expression is

**eLife digest** The neurons that transmit information around the nervous system develop in several stages. Embryonic stem cells specialize to form neuronal progenitor cells, which then develop into neurons. These cell types have different characteristics, in part because they make different proteins or different versions of the same proteins.

To make a protein, the DNA sequence of a gene is used to build a molecule of ribonucleic acid (RNA) that acts as a template for the protein. However, not all of this sequence codes for the protein. The non-coding regions must be removed from the RNA, and the remaining "exons" joined together to form the final "mRNA" template. Not all of the exons are necessarily included in the final mRNA molecule. By joining together different combinations of exons, several different versions of a protein can be produced from a single gene. This process is known as alternative splicing.

One way that alternative splicing is controlled is through proteins that bind to RNA and determine which exons are included or excluded from the final mRNA molecule. PTBP1 is an RNA-binding protein that controls alternative splicing in embryonic stem cells and neuronal progenitor cells. Embryonic stem cells have the ability to develop into all the cells of the body. In contrast, neuronal progenitor cells are restricted in their development and only give rise to specialized cells of the nervous system. The role of PTBP1 in these properties was not clear.

Linares et al. have now used a range of techniques to study the RNA molecules produced in these two cell types and how these RNAs change when PTBP1 is removed. This identified many RNAs whose splicing is regulated by PTBP1, including mRNAs of the gene that produces a protein called Pbx1, which is an important regulator of neuronal development.

Further investigation revealed that PTBP1 prevents a particular exon being included in the mRNA template for Pbx1. This creates an embryonic stem cell form of Pbx1 that does not affect neuronal genes. Removal of PTBP1 allows splicing of the Pbx1 exon and produces a version of Pbx1 that is found in neuronal progenitor cells and which turns on neuronal genes. Thus, through its action on Pbx1, one role of PTBP1 is to enable stem cells to maintain their non-neuronal properties and prevent their premature development into neuronal progenitor cells.

The gene for Pbx1 is only one of many genes controlled by PTBP1 at the level of splicing. One challenge for the future will be to understand how these genes work together in a common program that determines the properties of stem cells. Another question regards how the different Pbx1 proteins in stem cells and in neuronal progenitors can exert different effects in the cells where they are made.

repressed allowing the induction of PTBP2 and the initiation of a neuronal splicing program that is essential for neuronal maturation and survival (*Boutz et al., 2007*; *Li et al., 2014*; *Licatalosi et al., 2012*; *Makeyev et al., 2007*; *Tang et al., 2011*; *Zheng et al., 2012*; *Gueroussov et al., 2015c*). In addition to splicing, the PTB proteins affect other aspects of posttranscriptional regulation in neurons, including miRNA targeting (*Xue et al., 2013*; *Yap et al., 2012*). It has been shown that depletion of PTBP1 from fibroblasts is sufficient to drive cells towards a neuronal phenotype (*Xue et al., 2013*). As neurons mature, PTBP2 expression is eventually reduced, giving rise to an adult neuronal splicing program (*Li et al., 2014*; *Licatalosi et al., 2012*; *Tang et al., 2011*; *Zheng et al., 2012*). Together, these transitions in PTBP1 and PTBP2 expression define three phases of neuronal maturation each with different splicing programs.

PTBP1 and PTBP2 are encoded on separate genes and have very similar sequences and RNA-binding properties (*Markovtsov et al., 2000*; *Oberstrass et al., 2005*). These proteins target overlapping but not identical sets of splicing events (*Boutz et al., 2007*; *Li et al., 2014*; *Llorian et al., 2010*; *Tang et al., 2011*; *Zheng et al., 2012*). Some exons are responsive to both proteins (*Boutz et al., 2007*; *Li et al., 2014*; *Spellman et al., 2007*; *Zheng et al., 2012*). In these cases, PTBP1 and PTBP2 serve to repress adult splicing patterns during neuronal maturation, yielding isoforms that are expressed only after PTBP2 levels have declined. In other cases, exons are more sensitive to PTBP1 and shift their splicing when PTBP1 is shut off early in neuronal differentiation even though PTBP2 is present (*Boutz et al., 2007*; *Makeyev et al., 2007*; *Markovtsov et al., 2000*;

*Tang et al., 2011*). PTBP1 is known to regulate splicing in a wide variety of cell-types, and much of the work on PTBP1 targeting has used non-neuronal cell lines such as HeLa cells (*Llorian et al., 2010*; *Spellman et al., 2007*; *Xue et al., 2009*; *Xue et al., 2013*). The role of the PTBP1-specific splicing program in early progression along the lineage to neuronal progenitor cells has not been examined.

Directed differentiation of embryonic stem cells (ESCs) provides a versatile model for the study of neuronal commitment, differentiation, and maturation. These systems have illuminated a wide range of genetic contributors to neuronal phenotype including epigenetic modifiers, transcription factors, miRNAs, and signaling molecules (*Temple, 2001*; *Louvi and Artavanis-Tsakonas, 2006a*; *Kosik, 2006b*; *Hirabayashi and Gotoh, 2010*; *Ronan et al., 2013a*). Here we use mouse ESC culture to characterize the PTBP1 splicing program during early neuronal differentiation. We identify a diverse set of splicing events regulated by PTBP1, including an alternative exon in the homeodomain transcription factor Pbx1, whose switch in splicing induces a neuronal transcriptional program early in neuronal development.

## Results

### The transition in PTB protein expression occurs during in vitro neuronal differentiation

To examine PTBP1 and PTBP2 expression in an in vitro model of neuronal development, we differentiated mouse embryonic stem cells (ESCs) into motor neurons (MNs). Mouse ESCs (Day -2) were grown in aggregate culture for two days to form embryoid bodies (EBs; Day 0), which were then treated with retinoic acid (RA) and a Sonic hedgehog (Shh) pathway agonist for 5 days to induce MN formation (*Wichterle et al., 2002*; *Adams et al., 2015d*). To facilitate MN identification and isolation, we used a mouse ESC line that expresses eGFP under the control of the MN specific HB9 promoter (*Wichterle et al., 2002*). At Day 5, the GFP+ embryoid bodies were dissociated and MNs plated onto Matrigel in the presence of neurotrophic factors to permit further MN maturation (*Figure 1A*).

Similar to observations in the brain, as ESCs differentiate into MNs, PTBP1 expression declines, while PTBP2 expression is induced (*Figure 1B and C*) (*Boutz et al., 2007*; *Tang et al., 2011*; *Zheng et al., 2012*). By immunofluorescence, Day 2 MNs express the progenitor marker Nestin, whereas Day 5 GFP+ cells express the postmitotic MN marker HB9 and the neuronal marker TuJ1 (*Figure 1—figure supplement 1A and B*, and data not shown). The presence of PTBP2 in GFP+ cells was also seen by immunofluorescence (*Figure 1—figure supplement 1C*). To monitor PTBP1 and PTBP2 protein specifically in the differentiated neurons, we used fluorescence-activated cell sorting (FACS) to isolate GFP+ cells at Days 5 and 8. By western blot, these GFP positive cells express low amounts of PTBP1 and high levels of PTBP2 at Day 5. At Day 8 only trace amounts of PTBP1 are found, while PTBP2 expression remains high (*Figure 1—figure supplement 1D*). We did not observe the late decline in PTBP2 expression that occurs during neuronal maturation in vivo. This decline in PTBP2 expression may take place with longer culture times, or cells may not fully mature in this system. These results show that the switch from PTBP1 to PTBP2 expression occurs during MN differentiation, similar to earlier observations in the developing cortex.

We previously observed high levels of PTBP1 and low levels of PTBP2 in neuronal progenitor cells (NPCs) (*Boutz et al., 2007*; *Li et al., 2014*; *Licatalosi et al., 2012*). Nestin+ NPCs are observed early in EB cultures, but these cells could not be isolated in pure populations (*Figure 1—figure supplement 1A*). To examine NPCs in more detail, ESCs were grown in N2B27 media and expanded in the presence of EGF and FGF, to yield cultures where greater than 80% of the cells were Nestin+ and with very few TuJ1+ neurons (*Conti et al., 2005*; *Ying et al., 2003*) (*Figure 1—figure supplement 1E and F*). We examined both the Hb9-GFP line and the 46C ESC line used in the original monolayer NPC protocol, finding minimal differences in marker staining (*Ying et al., 2003*). By western blot, we found that both PTBP1 and PTBP2 proteins were expressed in these NPC cultures. As these NPCs were differentiated into neurons with retinoic acid, PTBP1 protein expression was lost while PTBP2 protein was induced (*Figure 1—figure supplement 1G and H*). The EB and monolayer culture systems likely give rise to different neuronal populations with differences in gene expression attributable to different neuronal types (*Gaspard et al., 2009*; *Wichterle et al., 2002*; *Lee et al., 2000b*;

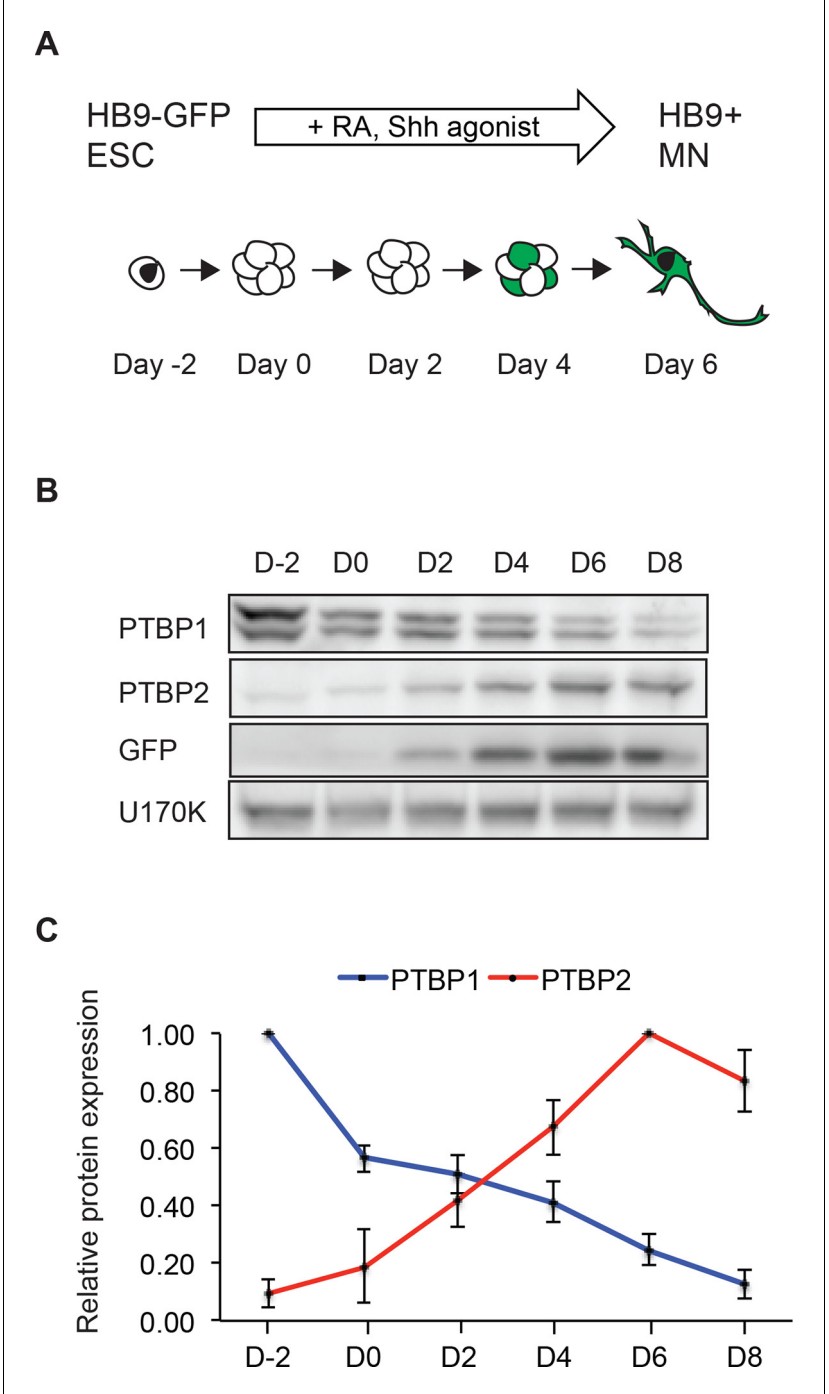

**Figure 1.** The transition in PTB protein expression occurs during in vitro neuronal differentiation. (**A**) HB9-GFP ESCs express eGFP under the MN-specific promoter HB9. The addition of retinoic acid (RA) and a Sonic hedgehog (Shh) agonist drives MN differentiation. (**B**) Western blot shows the loss of PTBP1 protein and gain of PTBP2 and GFP proteins as ESCs differentiate into HB9+ MNs. U170K served as a loading control. (**C**) Quantification of relative PTBP1 and PTBP2 protein expression across MN differentiation. Error bars represent standard error of the mean (SEM, n=2).

The following source data and figure supplements are available for figure 1:

**Source data 1.** Splicing changes identified by RNA-seq during ESC neuronal differentiation.

**Figure supplement 1.** Transitions in PTB protein expression as ESCs differentiate into MNs, NPCs, and neurons.

*Figure 1 continued on next page*

*Figure 1 continued*

**Figure supplement 2.** Transitions in alternative splicing occur during neuronal differentiation.

*Murashov et al., 2005a*; *Su et al., 2006c*; *Salero and Hatten, 2007*). Nevertheless, the transition in PTBP1-PTBP2 protein expression is common to both models of in vitro neuronal development, as well as embryonic brain.

We next examined alternative splicing patterns under different states of PTBP1 and PTBP2 expression. Poly-A plus RNA was isolated from undifferentiated ESCs (high PTBP1 and low PTBP2), Nestin+ progenitor cells (intermediate PTBP1 and PTBP2), and post-mitotic HB9-GFP+ MNs (low PTBP1 and high PTBP2). Strand specific libraries for each cell type were subjected to 100 nt paired-end sequencing to generate 200 to 250 million mapped reads per sample. Alternative splicing events were quantified using SpliceTrap to identify cassette exons, 5' and 3' splice sites, and retained introns that differ between all sample pairs (*Wu et al., 2011*). PSI values for the HB9 and 46C cell lines were highly correlated ($R^2$ = 0.92 and 0.90 in ESCs and NPCs respectively), indicating that the two cell lines are similar in splicing phenotype (data not shown). Differentially spliced events were filtered for p-value below 0.05 and ranked by the change in percent spliced in value (ΔPSI). Using a ΔPSI cutoff of 15%, we identified a large set of alternative splicing events that change across neuronal differentiation (*Figure 1—figure supplement 2A and B*). With these filters 1201 differential splicing events were identified between ESCs and MNs, 1218 differential events between NPCs and MNs, and 750 differential events between ESC and NPC (*Figure 1—source data 1*). To identify enriched functional categories, transcripts exhibiting differential splicing were compared to the total expressed transcript set by Gene Ontology analyses (*Huang et al., 2009a*). Differential splicing events between NPCs and MNs were enriched (FDR<0.05) in biological processes and cellular compartments relevant to neuronal differentiation such as cell projection organization, axonogenesis, and cell morphogenesis (*Figure 1—figure supplement 1C and D*). It is expected that a subset of these splicing changes are the result of changes in PTBP1 and PTBP2 expression.

## PTBP1 regulates a large set of neuronal exons in ESCs and NPCs

To identify neuronal splicing events controlled by the PTB proteins and in particular PTBP1, RNAi knockdowns were performed with two independent sets of control, PTBP1, and PTBP2 siRNAs. PTBP1 siRNA treatment depleted the protein by 80% (n=3) and 85% (n=2) in ESCs and NPCs respectively (*Figure 2A and B*). As seen previously, PTBP1 depletion induced PTBP2 protein expression by four-fold (*Boutz et al., 2007*; *Makeyev et al., 2007*; *Spellman et al., 2007*) (*Figure 2B*). This change in PTBP2 expression is comparable to the 3 and 13-fold increases in PTBP2 protein seen when ESCs differentiate into NPCs and Day 4 RA-derived neurons (*Figure 1—figure supplement 2G*). To assess the effect of PTBP2, we performed PTBP1/2 double knockdowns to remove 84% and 93% PTBP2 in ESCs and NPCs respectively (*Figure 2A and B*). 50 nt paired-end RNA sequencing was performed on the control and knockdown samples, and the splicing was analyzed with Splice-Trap. PSI values between the siRNA pairs showed a strong correlation ($R^2$ = 0.97 for each pair, data not shown), and the samples were pooled to increase read depth.

Using the *P*-value and PSI cutoffs previously described, we identified 418 splicing events altered by PTBP1 knockdown in ESCs, including 362 cassette exons, 50 alternative 5' or 3' splice sites, and 6 retained introns (*Figure 2—figure supplement 1B and C*, *Figure 2—source data 1*). This number was only slightly increased by double PTBP1/2 depletion, with similar changes in PSI when compared with the single knockdown (*Figure 2—figure supplement 1C*). The splicing changes between the single and double knockdowns showed a strong correlation ($R^2$ = 0.91, *Figure 2—figure supplement 1E*), indicating that these events are more strongly affected by PTBP1. A majority of the exons altered by PTB protein depletion also change in the same direction during ESC to MN differentiation (exons in the upper right and lower left quadrants of the plots in *Figure 2C*). Applying a ΔPSI cutoff of 15% for both data sets, 30% of neuronal exons are regulated by PTBP1, including 97 neuronally activated exons that are PTBP1 repressed, and 56 neuronally skipped exons that are PTBP1 activated (indicated by red and blue data points respectively in *Figure 2C*, left panel; *Figure 2—source data 2*). 31% of neuronal exons were PTBP1/2 repressed (101) and activated (55) following PTBP1/2

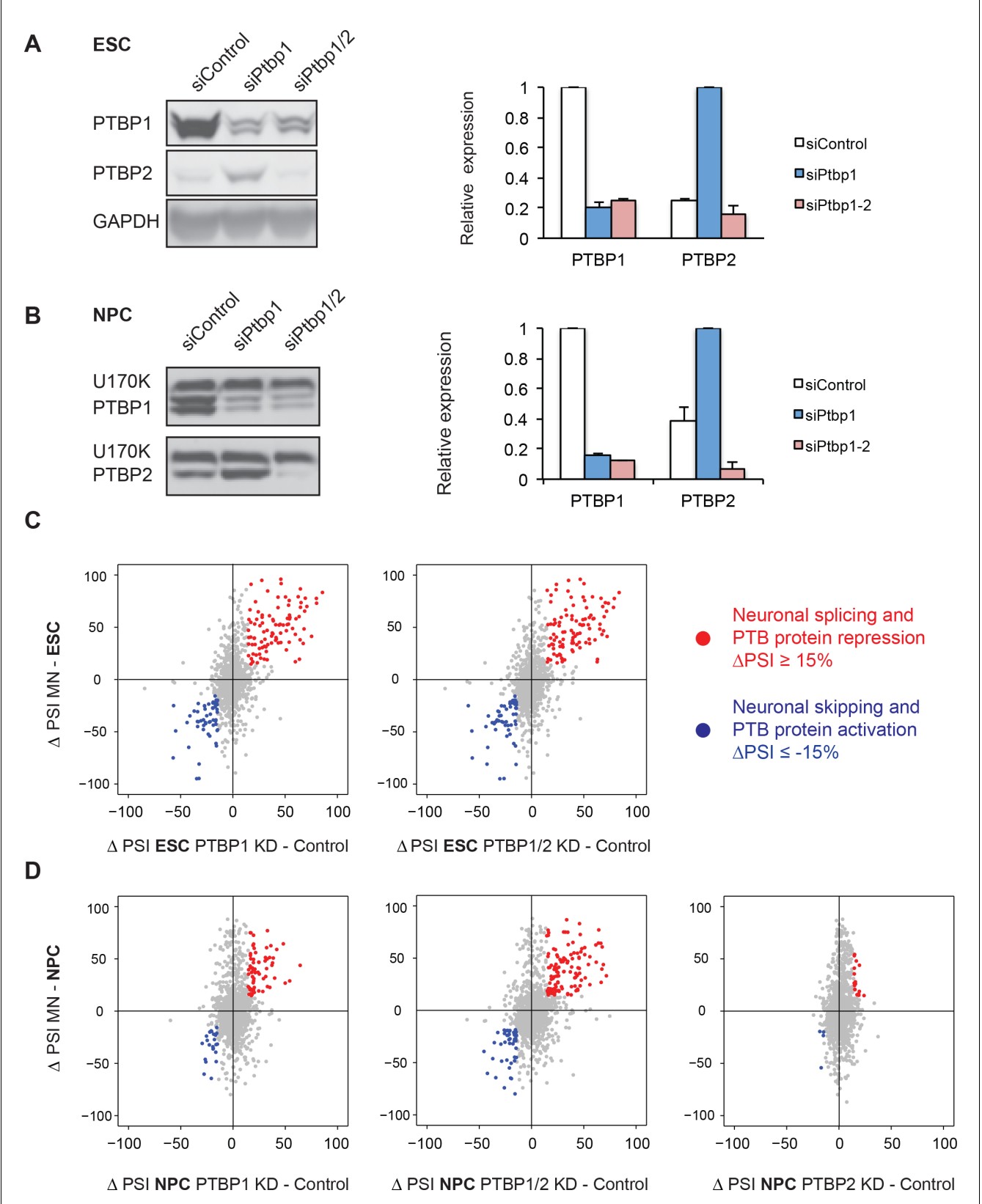

**Figure 2.** PTBP1 regulates a large set of neuronal exons in ESCs and NPCs. (**A, B**) Left: Western blots of ESCs and NPCs treated with siControl, siPtbp1, or both siPtbp1 and siPtbp2. Similar results were obtained with an independent set of siRNAs. Right: Bar graphs showing the relative PTBP1

*Figure 2 continued on next page*

*Figure 2 continued*

and PTBP2 protein expression ± SEM following siRNA treatment in ESCs (n=3) and NPCs (n=2). Cells with the highest level of either protein are normalized to 1. (C) Scatter plots compare the splicing changes in individual exons resulting from PTB protein depletion in ESCs (X-axis) with splicing changes between ESCs and MNs (Y-axis) ( *Figure 2—source data 2)*. (D) Scatter plots compare the splicing changes resulting from PTB protein depletion in NPCs (X-axis) with splicing changes between NPCs and MNs (Y-axis) ( *Figure 2—source data 4)*. (C, D) Data points in red correspond to neuronally spliced exons that are PTB protein repressed (ΔPSI ≥ 15%). Data points in blue correspond to neuronally skipped exons that are PTB protein activated (ΔPSI ≤ -15%). Data points in grey correspond to the other cassette exons measured by SpliceTrap.

The following source data and figure supplement are available for figure 2:

**Source data 1.** Splicing changes identified by RNA-seq following PTB protein depletion in ESCs.
**Source data 2.** Neuronal cassette exons regulated by the PTB proteins in ESCs.
**Source data 3.** Splicing changes identified by RNA-seq following PTB protein depletion in NPCs.
**Source data 4.** Neuronal cassette exons regulated by the PTB proteins in NPCs.
**Source data 5.** Cassette exons co-regulated by PTBP1 and PTBP2 in NPCs.
**Figure supplement 1.** The PTB proteins regulate a large set of exons in ESCs and NPCs.

double knockdown (*Figure 2C*, right panel). As expected, many exons that change during neuronal differentiation are not affected by PTB protein depletion, indicating likely regulation by other proteins (grey data points along the y-axes of the plots in *Figure 2C*).

PTBP1 depletion from NPCs altered 325 splicing events (264 cassette exons, 44 alternative 5' or 3' splice sites, and 17 retained introns, *Figure 2—figure supplement 1B*, *Figure 2—source data 3*). Comparing this list to the splicing changes between NPCs and Day 5 GFP+ MNs identified 16% of neuronal exons regulated by PTBP1 (66 PTBP1-repressed exons increased with differentiation; 23 PTBP1-activated exons decreased with differentiation; *Figure 2D*, left panel; *Figure 2—source data 4*). Interestingly, PTBP1/2 double knockdown increased the number of regulated neuronal exons to 30% (116 PTBP1/2-repressed exons increased with differentiation; 45 PTBP1/2-activated exons decreased with differentiation; *Figure 2D*, middle panel). These changes in splicing after double PTBP1/2 knockdown in NPCs also tended to be larger than PTBP1 knockdown alone (*Figure 2—figure supplement 1D*), yielding a lower correlation than seen in ESC (R$^2$ = 0.69, *Figure 2—figure supplement 1F*). Knockdown of PTBP2 alone in NPCs had only a small effect on the splicing of neuronal exons, indicating that PTBP2 largely regulates a subset of PTBP1 targets, with relatively few targets of its own (*Figure 2—figure supplement 1A and D*) (*Figure 2D*, right panel). This is in agreement with previous results that a large set of exons in the developing mouse brain is regulated by both PTBP1 and PTBP2. These include Dlg4 (Psd-95) exon 18 (*Boutz et al., 2007*; *Li et al., 2014*; *Zheng et al., 2012*). In these new data, PTBP1 and PTBP2 single knockdown each only moderately increased Dlg4 exon 18 splicing (ΔPSI = 8% and 13% respectively), whereas PTBP1/2 double knockdown increased exon 18 splicing by 42%. Other exons where PTBP2 can compensate for the loss of PTBP1 are found in the Magi1, Ap2a1, and Tnrc6a transcripts (*Figure 2—source data 5*). Thus, PTBP1 and PTBP1/2 depletion from NPCs identifies exons that are coregulated by both proteins. Whereas, depletion from ESCs identifies exons affected largely by PTBP1.

The PTBP target transcripts in ESCs and NPCs include both unique and overlapping splicing events (80 shared, 73 unique to ESCs, 81 unique to NPCs). To examine possible common functions of the PTBP1/2 regulatory programs, Gene Ontology analyses were performed relative to the transcripts expressed in each cell type (*Huang et al., 2009*). These yielded fewer enriched terms compared to total set of neuronally regulated exons (*Figure 1—figure supplement 2C and D*). Neuronal exons regulated by PTBP1 in ESCs showed significant enrichment (FDR = 0.04) in the cellular compartment ontology term, cell junction. Neuronal exons regulated by PTBP1/2 in NPCs were enriched (FDR = 0.01) in the cellular compartment ontology term, cytoskeleton. The total set of neuronal exons identified during ESC and NPC differentiation were also enriched for these terms among others (*Figure 1—figure supplement 2C and D*). The relative lack of enriched terms might reflect a

broad number of functions for PTBP targets, but this will require further analysis to assess. Enrichment for cytoskeletal functions is in agreement with previous analyses of PTBP2 function in mice, and with the dramatic cytoskeletal changes during the transition from PTBP1 to PTBP2 during early neuron differentiation (*Li et al., 2014*; *Licatalosi et al., 2012*).

## PTBP1 iCLIP-seq identifies direct PTBP1 splicing targets during neuronal differentiation

To identify direct binding of PTBP1 to transcripts in these cells, we performed crosslinking-immunoprecipitation followed by sequencing (iCLIP-seq) (*König et al., 2010*; *König et al., 2011*). UV irradiated ESC and NPC cultures were immunoprecipitated with PTBP1 antibody or with Flag antibody as a negative control (*Figure 3—figure supplement 1A*). Cross-linked RNA fragments were converted into cDNA using a modification of the iCLIP protocol and subjected to high-density sequencing (*König et al., 2010*). After removing PCR duplicates, unique sequencing reads were mapped to the UCSC known Genes table (*Hsu et al., 2006d*). To identify significant crosslinking sites, the FDR for each position was calculated (*König et al., 2010*). Applying an FDR cutoff of 0.01, we obtained 918,725 and 914,084 significant reads of RNA crosslinked to PTBP1 in ESCs and NPCs respectively.

The PTBP1-bound sequences were analyzed for base content and location. The iCLIP protocol generally yields fragments that terminate one nucleotide downstream of the crosslink site (*König et al., 2010*). We compiled a set of binding regions extending 20 nucleotides upstream and downstream of a crosslink site. We measured the frequency of all pentamer motifs within these binding sites, relative to randomly chosen intervals from the same introns. As expected, PTBP1 binding sites in both ESC and NPC were highly enriched for CU-rich pentamers. The top 20 motifs included pentamers comprised of only C and U nucleotides, as well as some with G nucleotides, in agreement with the described binding specificity of PTBP1 (*Figure 3—figure supplement 1B*) (*Amir-Ahmady et al., 2005*; *Han et al., 2014*; *Licatalosi et al., 2012*; *Llorian et al., 2010*; *Oberstrass et al., 2005*). The enriched motifs were very similar between ESC and NPC samples ($R^2$ = 0.93, *Figure 3—figure supplement 1B*). The locations of the binding sites were also consistent with previous observations, with most being found in introns and some in 3' UTRs (*Figure 3—figure supplement 1C and D*) (*Xue et al., 2009*).

We next examined the alternative exons sensitive to PTBP1 knockdown ($\Delta$PSI $\geq$ 15%) for PTBP1 iCLIP clusters within 500 nt upstream or downstream ($\geq$ four significant reads per cluster). In ESCs, we identified 170 PTBP1-dependent exons that are bound by PTBP1 (*Figure 3—figure supplement 1E*, *Figure 3—source data 1*). In NPCs, we identified 108 PTBP1-dependent exons with PTBP1 binding (*Figure 3—figure supplement 1F*, *Figure 3—source data 1*). These include many known PTBP1 target transcripts such as Ganab, Rod1, Tpm1, Rbm27, Fam38a, Mink1, and Ptbp2 (*Boutz et al., 2007*; *Makeyev et al., 2007*; *Spellman et al., 2007*; *Xue et al., 2009*). Some known target exons with PTBP1 binding did not exhibit splicing changes above our cutoff, including Pkm2, Eif4g2, Dlg4, and Ptbp1 (*Xue et al., 2009*; *Zheng et al., 2013*). These exons may be controlled by additional factors in ESCs and NPCs.

Overlapping all of the datasets, we defined a minimal set of 104 direct PTBP1 targets in ESCs and NPCs, whose splicing changes during neuronal differentiation (70 Ptbp1 repressed, 34 Ptbp1 activated; *Figure 3—source data 2*). Many other exons are expected to be direct targets but were excluded due to failure to fulfill a cutoff value in one dataset. The exons in this minimal set affect a wide range of cellular processes including GTPase signaling (Dock7), synaptic function (Gabbr1), and transcriptional regulation (Tcf20) (*Figure 3A–D*). For example, Gabbr1 is a subunit of a metabotropic gamma-aminobutyric acid (GABA) receptor. PTBP1 induced skipping of Gabbr1 exon 15 leads to premature translation termination in exon 16 and predicted nonsense mediated decay of the Gabbr1 mRNA (*Makeyev et al., 2007*). We observe PTBP1 binding within exon 15 and upstream, consistent with the increased exon 15 splicing observed during neuronal differentiation and after PTBP1 depletion (*Figure 3A*). This gene is similar to Dlg4, where PTBP1 regulates its overall expression during development rather than the production of a neuronal isoform. However unlike Dlg4, Gabbr1 is more sensitive to PTBP1 than PTBP2, allowing its expression to be induced earlier.

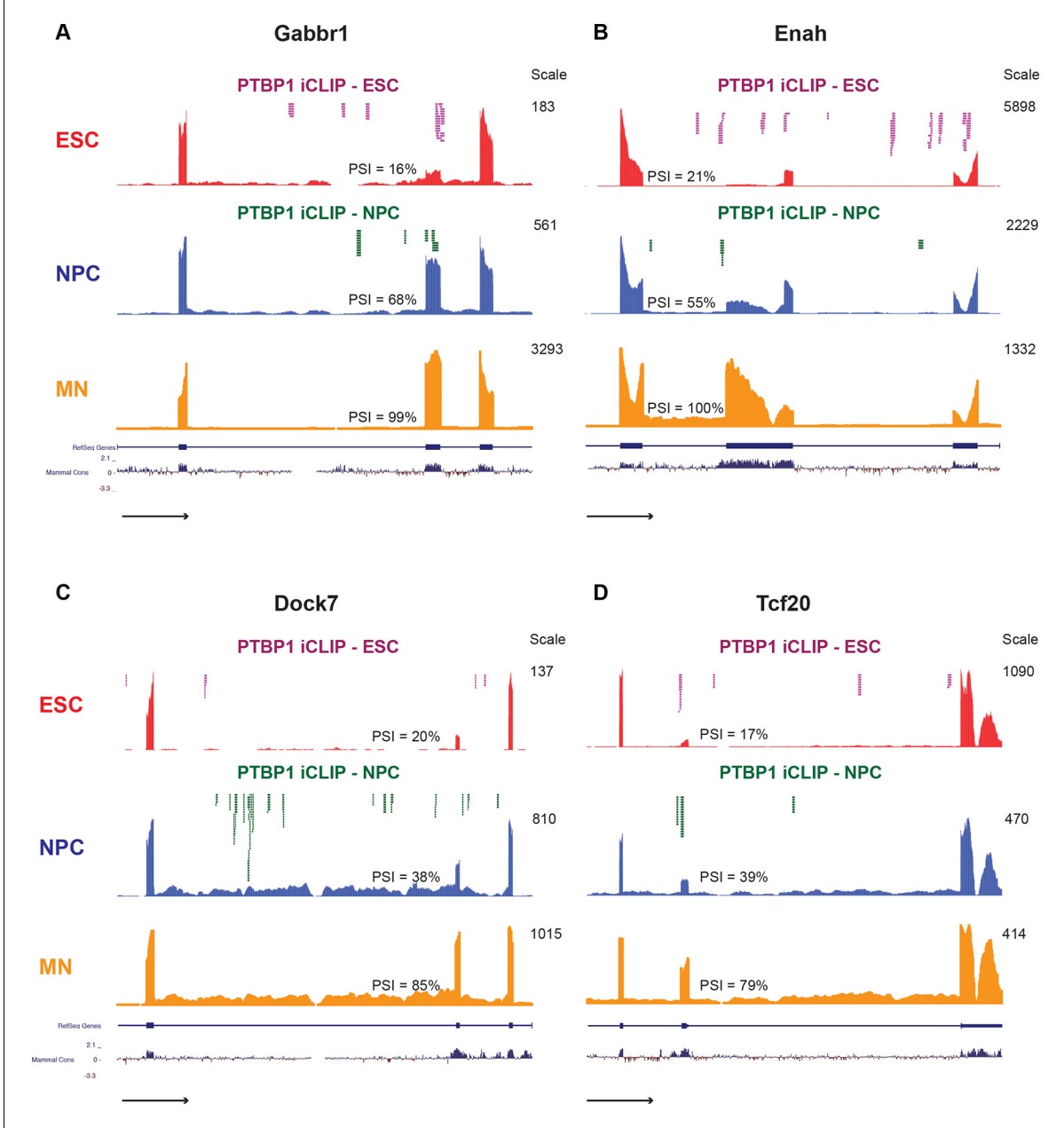

**Figure 3.** Examples of neuronally spliced cassette exons that are PTBP1 repressed. (**A-D**) Genome Browser tracks show aligned RNA-seq reads from ESCs (red), NPCs (blue), and GFP+ MNs (yellow). The scale indicates the number of mapped reads in the highest peak. PSI values for the cassette exons were calculated with SpliceTrap (*Wu et al., 2011*). Significant PTBP1 iCLIP sequencing reads from ESCs (magenta) and NPCs (green) are overlaid on the Genome Browser tracks.

The following source data and figure supplement are available for figure 3:

**Source data 1.** PTBP1-regulated cassette exons with PTBP1 binding.

**Source data 2.** Direct PTBP1 target exons during neuronal differentiation.

**Figure supplement 1.** The base content and location of PTBP1 iCLIP-sequencing reads are consistent with known PTBP1 binding properties.

## PTBP1 regulates a switch in Pbx1 isoform expression

The above data define the PTBP1 regulatory program in a new level of detail, and show that for early cells along the neuronal lineage, the PTBP1 program is distinct from that regulated by PTBP2. PTBP1 target transcripts influence an array of important cellular functions, and understanding how splicing alters each target's activity in differentiating cells will be a challenge. We were particularly interested in how the early PTBP1 program affects gene regulation. We identified a number of transcriptional regulators whose splicing is controlled by PTBP1 during ESC differentiation into NPCs, including Tcf20, Med23, Gatad2a, and Pbx1. These early changes in the structure of transcriptional regulators have the potential to broadly alter gene expression programs during neuronal development (*Gabut et al., 2011*; *Raj et al., 2011*; *Demir and Dickson, 2005b*).

We focused on the homeodomain transcription factor Pbx1 because of its known interactions with the Hox family of developmental regulators (*Piper et al., 1999a*; *LaRonde-LeBlanc, 2003a*). Multiple studies implicate Pbx1 and its family members as transcriptional regulators during neuronal development (*Maeda et al., 2002a*; *Vitobello et al., 2011b*; *Sgadò et al., 2012c*; *Schulte et al., 2014*). Exon 7 in the Pbx1 transcript is conserved across mammals and its inclusion leads to production of the Pbx1a isoform. Skipping exon 7 shifts the translational reading frame to introduce a new termination codon (PTC) in exon 8 (*Figure 4A*). This translation stop does not result in NMD but instead generates the shorter isoform, Pbx1b, which retains the DNA binding homeodomain but lacks the 83 amino acids at the C-terminus of Pbx1a. During development, early embryonic tissues predominantly express Pbx1b, whereas Pbx1a is found in neural tissues (*Redmond et al., 1996*; *Schnabel et al., 2001b*). We found a similar isoform switch in cultured cells, where ESCs express low levels of Pbx1b. As ESCs differentiate into NPCs and HB9+ MNs, both overall Pbx1 expression and exon 7 splicing are induced, such that MNs predominantly express the Pbx1a isoform (*Figure 4B and C*). Previous work indicates that the Pbx1a and Pbx1b isoforms have different transcriptional activities and cellular functions. The proteins differ in their ability to activate or repress reporter gene expression, their interactions with the transcriptional corepressors NCoR1 and NCoR2, and their activity for cell transformation when fused to E2A (*Kamps et al., 1991*; *Di Rocco et al., 1997*; *Asahara et al., 1999b*). However, the role of these functional differences during neuronal differentiation is not understood.

We find that Pbx1 exon 7 is regulated by PTBP1. Exon 7 is derepressed as ESCs differentiate into NPCs and further into motor neurons (*Figure 4B and C*). Depletion of PTBP1 from ESCs or NPCs strongly induces exon 7 splicing (*Figure 4D and E*). In NPCs, a PTBP1 iCLIP cluster is found in the intron upstream of exon 7 (*Figure 4B*). These iCLIP tags are not seen in ESCs presumably due to lower Pbx1 transcript levels in these cells (*Figure 4B and C*). These data indicate that one consequence of PTBP1 depletion during neuronal differentiation is to switch Pbx1 expression from the Pbx1b isoform to Pbx1a.

## Intron 6 deletions in Pbx1 induce exon 7 splicing in ESC

PTBP1 represses the production of Pbx1a in ESCs and during early neuronal differentiation. To examine the role of Pbx1a repression, we modified the Pbx1 locus using the CRISPR-Cas9 technology to force exon 7 splicing in cells containing PTBP1 (*Doudna and Charpentier, 2014*; *Cong and Zhang, 2015e*). Intron 6 upstream of exon 7 contains a PTBP1 iCLIP cluster as well as regions of mammalian sequence conservation, possibly indicative of additional splicing regulatory elements. Guide RNA pairs were designed to target the Cas9 endonuclease to sites within intron 6 (*Figure 5A*). After Cas9 catalyzes double-strand breaks at each site, non-homologous end joining (NHEJ) of the free ends will potentially generate genomic deletions within intron 6 (*Figure 5—figure supplement 1A*). The largest deletion, between guides 1 and 4, would remove 5.5 kb of intron 6 containing all of the conserved regions, while preserving the splice sites (*Figure 5A*). HB9-GFP ESCs were transiently transfected with expression plasmids for Cas9 and two guide RNAs, and clones were isolated from single transfected cells and genotyped by PCR, with the deletion event confirmed by sequencing. Using guides g1 and g4, we identified no homozygous deletions among 60 ESC clones, although heterozygotes were readily isolated (*Figure 5—figure supplement 1B and C*). We expanded five heterozygous deletion clones with preserved splice sites. Notably, we were still unable to isolate homozygous null cells after reintroducing guides 1 and 4 into a heterozygous cell line. Shorter deletions were also tested. Guides 3 and 4 generated a 1.4 kb deletion that includes

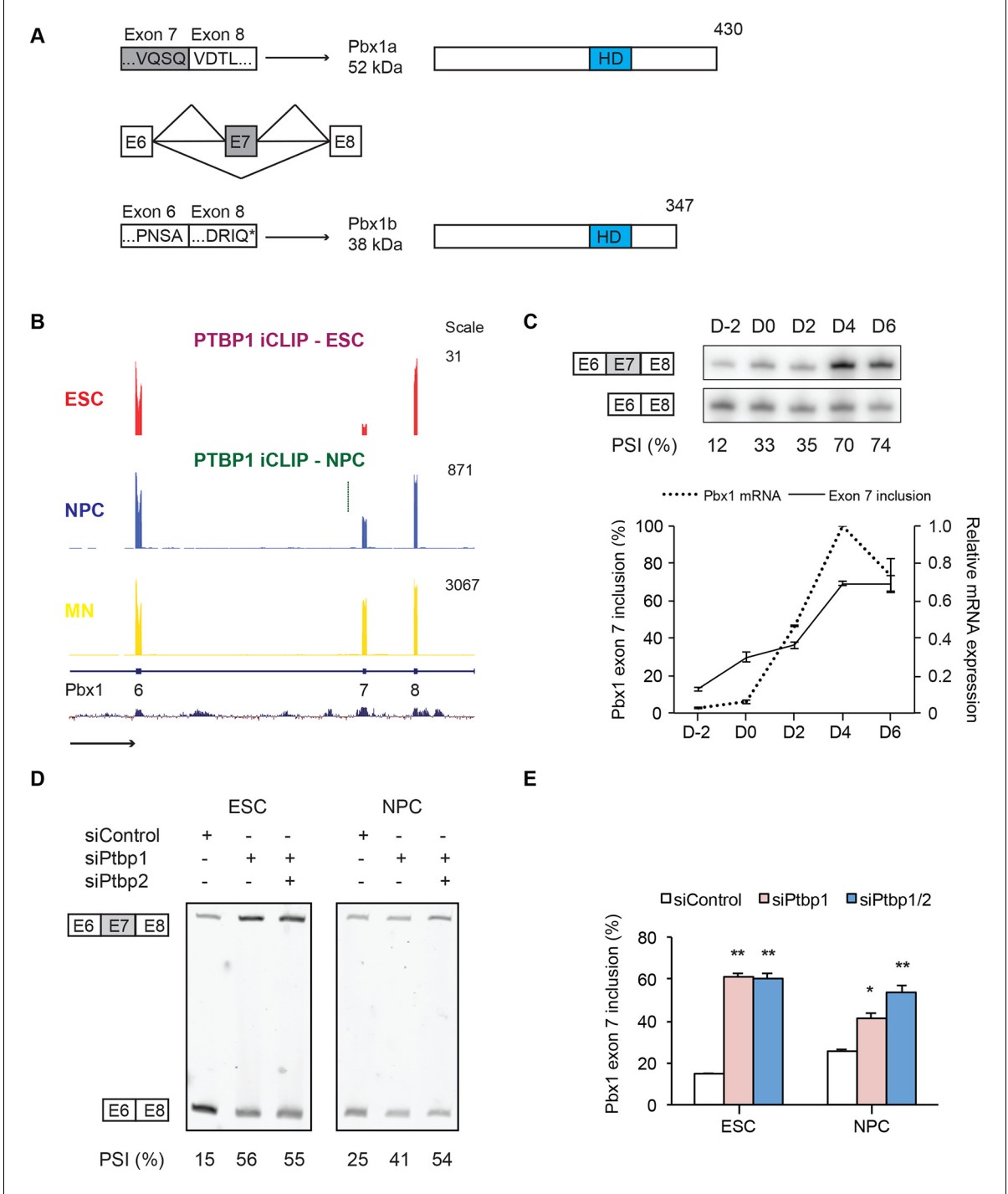

**Figure 4.** PTBP1 regulates a switch in Pbx1 isoform expression. (**A**) Exons are represented as boxes, while horizontal lines represent introns. Inclusion of exon 7 results in the longer Pbx1a protein (top). Skipping of exon 7 results in a frameshift and translation termination at a stop codon (*) in exon 8 to yield the shorter Pbx1b protein isoform (bottom). The DNA-binding homeodomain (HD) is indicated by a blue box. (**B**) Genome Browser tracks of aligned RNA-seq reads show exon 7 inclusion as ESCs (red) differentiate into NPCs (blue) and GFP+ MNs (yellow). A significant PTBP1 iCLIP cluster is present upstream of exon 7 in NPCs (green). (**C**) Both Pbx1 exon 7 (top panel, solid line bottom panel, n=3) and Pbx1 mRNA expression (dashed line bottom panel, n=2) are induced with MN differentiation. (**D**, **E**) ESCs and NPCs were treated with siControl, siPtbp1, or both siPtbp1 and siPtbp2. (**D**) RT-PCR of Pbx1 exon 7 splicing following siRNA treatment in ESCs (24 PCR cycles) and NPCs (19 PCR cycles). (**E**) Bar chart of Pbx1 exon 7 splicing (Mean ± SEM, n=3). Statistical analyses were performed using paired one-tailed Student's t-test (*P*-value<0.01**, 0.05*).

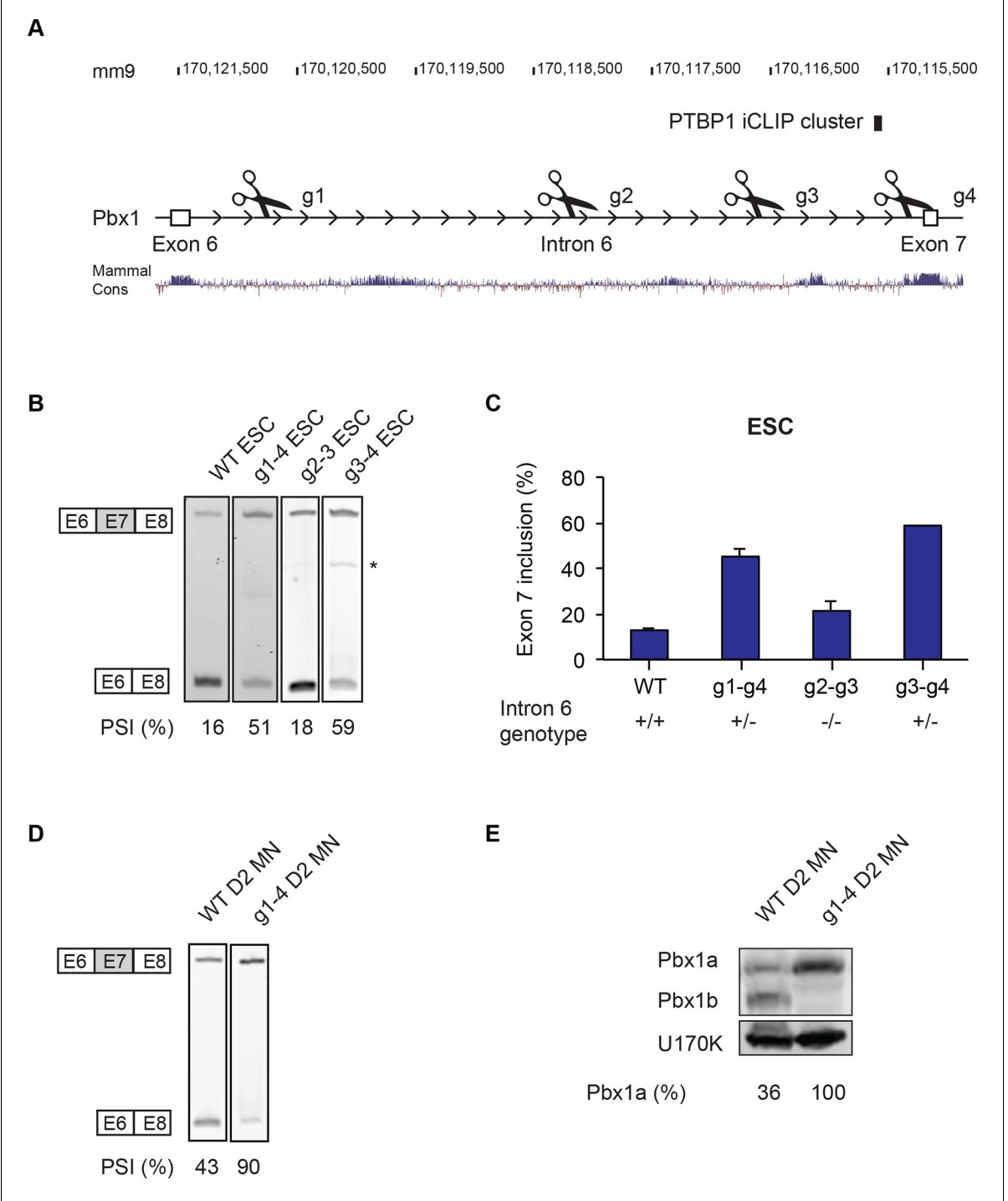

**Figure 5.** Intron 6 deletions in Pbx1 induce exon 7 splicing. (A) Guide RNAs were designed to target regions along Pbx1 intron 6. The location of the PTBP1 iCLIP cluster is indicated by the black box. (B) RT-PCR of Pbx1 exon 7 splicing in ESC clones carrying Cas9-mediated deletions. The asterisk (*) indicates a sporadic non-specific band. (C) Bar chart of Pbx1 exon 7 splicing in ESC clones with intron 6 deletions. Error bars indicate the SEM for three independent clones, except for g3-4 where only one clone was isolated. (D) RT-PCR of Pbx1 exon 7 splicing and (E) western blot of Pbx1 in D2 MN cultures show that Pbx1a is the predominant isoform in the I6 +/- cell line at day 2. Solid borders of RT-PCR gels indicate non-adjacent gel lanes.

The following figure supplement is available for figure 5:

**Figure supplement 1.** Cas9-targeted deletions of Pbx1 intron 6 switch Pbx1 isoform expression.

the PTBP1 binding site. Again this deletion was only isolated as a heterozygote (1/6 clones). In contrast, guides 2 and 3 generated a 1.5 kb deletion from the middle of the intron that was isolated as a homozygous allele (3/10 clones).

Interestingly, these deletions had different effects on Pbx1 splicing. Clones carrying the heterozygous Intron 6 g1-g4 deletion (I6 +/-) exhibited dramatically increased exon 7 splicing (*Figure 5B and C*). Exon 7 was included in only 16% of the Pbx1 mRNA in clones of the parental ESC line, whereas exon 7 was included in 51% of the mRNA in the I6 +/- clones. This indicates that RNA from the mutant allele is almost entirely splicing in exon 7 and confirms that intron 6 contains splicing silencer elements for this exon.

We also tested the other deletions within intron 6 for their effects on exon 7 splicing. The deletion derived from guides 3 and 4 was similar to the long deletion using guides 1 and 4, with the heterozygous deletion clone exhibiting strongly induced exon 7 splicing (g3-4 in *Figure 5B and C*). In contrast, the homozygous intronic deletion derived from guides 2 and 3 had minimal effect, with exon 7 splicing similar to wild type in 3 separate clones (g2-3 in *Figure 5B and C*). The inability to isolate cells carrying homozygous mutations that increase exon 7 splicing suggests that either the gain of Pbx1a or the loss of Pbx1b is deleterious for ESC growth.

## The early expression of Pbx1a promotes neuronal gene expression

To characterize how early Pbx1a expression alters transcriptional programs, we measured steady-state transcript levels in cells with different Pbx1 genotypes by RNA-seq. Wild type (n=3) and I6 +/- (n=5) ESC clones were differentiated into MNs. At Day 2, the I6 +/- cells predominantly expressed the Pbx1a mRNA isoform, leading to dramatically increased Pbx1a protein compared to wildtype cells (*Figure 5D and E*). These Day 2 samples were subjected to 50 nt single-end RNA sequencing, with the data analyzed by the Cufflinks pipeline. We performed hierarchical clustering on the replicates, and as expected the gene expression profiles were similar across replicates, with the profiles of the wild type and Pbx1 mutant cell replicates segregating into two groups (data not shown).

Using a twofold change in FPKM as the cutoff, we identified 33 differentially expressed genes in the I6 +/- cultures (31 induced, 2 repressed). Examining these genes with the largest changes in expression confirmed that a Pbx1a neuronal regulatory program was activated in the mutant cells. Of the 20 transcripts that are most altered in I6 +/-, 14 are also induced as wildtype ESCs differentiate in NPCs and MNs (*Figure 6—source data 1* and *2*). Multiple genes of this group are known to play important roles in neuronal differentiation including Phox2b, Cntn2, Ntng2, Olig1, Isl1, Nrp2, Ngfr, Nav2, and Slit1 (*Figure 6B*), and many were previously shown to be affected by Pbx1 (Igf2, Isl1, Dlk1, and Meox1) (*Kim et al., 2002*; *Jürgens et al., 2009b*; *Thiaville et al., 2012d*). Notably, the gene exhibiting the largest change in expression was Phox2b, a marker of hindbrain visceral motor neurons and a known Pbx1a target (*Pattyn et al., 2000c*; *Samad, 2004*).

To increase the number of differentially expressed genes, we relaxed the cutoff to a 1.5-fold change in expression and identified 196 induced and 25 repressed genes in the mutant relative to the wild type cells. Of the 196 induced genes, 139 increase expression with ESC neuronal differentiation. Gene ontology (GO) analyses were performed on the induced gene set to assess functional enrichments relative to the total transcripts present in the D2 MN cultures. Within the biological processes ontology, several terms were significantly enriched (FDR<0.05) including axonogenesis, pattern specification, regulation of transcription, cell adhesion, cell motion, cell fate commitment, and heart development (*Figure 6C*). Notably, the induced genes include the MN markers Olig2, Lhx3, and Mnx1 (*Figure 6—source data 1*), again indicating that Pbx1a is activating a neuronal transcriptional program.

Overlapping the 196 induced events with a published Pbx1 ChIP data set identified 52 genes with nearby Pbx1 binding (*Penkov et al., 2013b*) (*Figure 6—source data 1*). Even though these binding events were identified in E11.5 embryos, 26.5% of induced transcripts in D2 MNs exhibit Pbx1 binding. This is compared to 14.8% of all expressed genes, constituting a significant enrichment of Pbx1 binding sites adjacent to induced transcripts ($p$=1.06e-5 by hypergeometric test).

We were particularly interested in the up-regulation of several transcription factors with known roles in neuronal differentiation, including the homeobox transcription factor Hoxc5 (*Figure 6D*). Hoxc5 affects several aspects of MN development and function, including the regulation of the phrenic motor neuron column (*Liu et al., 2001c*; *Dasen et al., 2003b*; *Philippidou et al., 2012e*). To confirm Pbx1 binding in D2 MN cultures, we performed ChIP-qPCR. All the genes tested, including

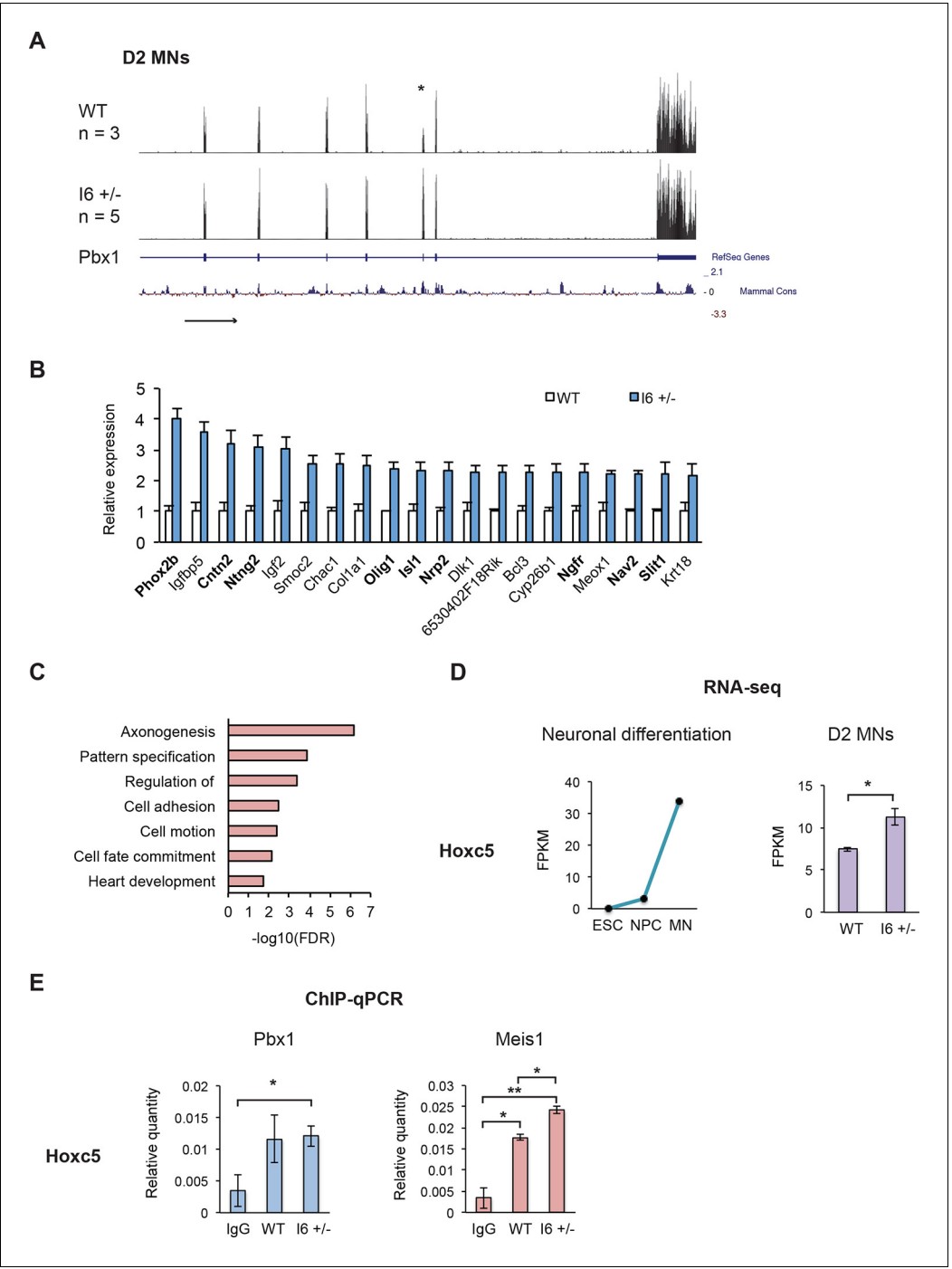

**Figure 6.** The early expression of Pbx1a promotes neuronal gene expression. (**A**) Genome Browser tracks of aligned RNA-seq reads from wild type (top) and I6 +/- (bottom) Day 2 MN cultures. The asterisk (*) indicates Pbx1 exon 7, which is induced in the mutant. (**B**) Bar charts of the top 20 induced genes in the I6 +/- cell lines compared to expression in wild type cells (*Figure 6—source data 1*). Genes highlighted in bold have known roles in neuronal differentiation. Error bars indicate SEM. (**C**) Gene ontology analysis of the 196 genes induced by 1.5 fold in the I6 +/- cultures. (**D**) Hoxc5 expression increases during neuronal differentiation (left panel, *Figure 6—source data 2*) and the induction of Pbx1a. (**E**) Relative Pbx1 (left panel) and Meis1 (right panel) binding in D2 MN cultures (n=3). Statistical analyses were performed using Welch's t-test (*P*-value<0.05*, <0.01**).

The following source data and figure supplement are available for figure 6:

**Source data 1.** Expression changes in I6 +/- Day 2 MN cultures.

*Figure 6 continued on next page*

*Figure 6 continued*

**Source data 2.** Expression values in ESCs, NPCs, and GFP+ MNs.

**Figure supplement 1.** Pbx1 binds target neuronal genes.

Hoxc5, Hoxa3, Midn, Tshz3, Zfp503, and Igf1r showed higher Pbx1 binding than an IgG control, although the significance relative to the control varied (n=3, *Figure 6E*, *Figure 6—figure supplement 1A–E*). Differences in Pbx1 binding were not observed between the wildtype and I6+/- cell lines, as might be expected since both splice variants maintain the DNA binding homeodomain of Pbx1.

Pbx1 is known to cooperate with the transcriptional cofactors Meis1 and Prep1 (Pknox1) in regulating gene expression. To assess whether Pbx1 isoforms differentially affect Meis1 and Prep1 recruitment, we performed ChIP-qPCR in D2 MN cultures at shared binding sites, identified in previous ChIP-seq studies (*Penkov et al., 2013b*). Most genes showed no change in Prep1 or Meis1 binding between wildtype and I6+/-, indicating that the Pbx1b and Pbx1a isoforms likely differ in other interactions. Notably, there was a significant increase in Meis1 binding at the Hoxc5 locus when Pbx1a was induced in I6+/- (*Figure 6E*). Thus, Pbx1a may increase Hoxc5 expression through enhanced recruitment of Meis1. Together the data indicate that switching to Pbx1a increases the expression of genes affecting neuronal fate, including Hoxc5.

## Discussion

Neuronal development is driven by a complex set of genetic events affecting every step in gene expression from transcriptional regulation by specialized DNA binding proteins to translational repression by miRNAs. These levels of regulation are intricately connected, where one type of regulator will target effectors at different steps in the gene expression pathway. In particular, alternative splicing plays a key role in defining the neuronal proteome with thousands of spliced isoforms found specifically in neurons or particular neuronal subtypes (*Barbosa-Morais et al., 2012*; *Grabowski and Black, 2001*; *Merkin et al., 2012*). A variety of RNA binding proteins alter splicing patterns during development and in the adult brain, including members of the PTBP, Rbfox, Nova, CELF, SRRM and other protein families (*Zheng and Black, 2013*; *Norris and Calarco, 2012a*; *Raj and Blencowe, 2015a*). The changes in isoform expression controlled by these factors determine the activity of many essential neuronal genes. However, these regulatory programs are incompletely characterized and the cellular roles of their alternatively spliced products are often not understood.

The polypyrimidine tract binding proteins, PTBP1 and PTBP2, influence neuronal differentiation through programs of posttranscriptional gene regulation. In previous work, we found that the transition from PTBP1 to PTBP2 expression during neuronal differentiation causes a large scale reprogramming of splicing patterns (*Boutz et al., 2007*). More recently, we identified a second transition late in neuronal maturation, where PTBP2 levels decline allowing for the expression of many adult-specific spliced isoforms (*Li et al., 2014*; *Licatalosi et al., 2012*; *Zheng et al., 2012*). Like many splicing regulators, the PTB proteins also affect other aspects of mRNA metabolism. The depletion of PTBP1 from mouse embryonic fibroblasts (MEFs) was shown to induce the trans-differentiation of these cells into neurons (*Xue et al., 2013*). This was linked to PTBP1 bound in 3′ UTRs altering miRNA targeting of transcripts driving neuronal differentiation, including the REST transcriptional repressor. In contrast, alterations in neuronal differentiation were not observed in mice carrying a conditional *Ptbp1* knockout mutation in nestin-positive NPCs (*Shibasaki et al., 2013*). Similarly, we have not found that PTBP1 depletion is sufficient to induce neuronal differentiation of ESCs. This could reflect a difference between ESCs and NPCs compared to MEFs, or differences in culture conditions. Notably, we find here that, like its 3′ UTR targets, the PTBP1 splicing program also includes targets that enhance neuronal development, such as Pbx1.

Earlier studies demonstrated that while PTBP1 and PTBP2 share many targets, exons can have different sensitivities to the two proteins (*Keppetipola et al., 2012*). This raised questions about the mechanisms of their differential targeting and the biological roles of the early and late neuronal

splicing programs. The late program was illuminated in the PTBP2 knockout mouse where many exons that are repressed by both proteins were identified (*Li et al., 2014*; *Licatalosi et al., 2012*). However, the early program composed of exons primarily targeted by PTBP1 was harder to define in the developing brain. By examining the PTBP1 splicing program in ESC culture, we now identify a large set of targets primarily responsive to PTBP1 regulation. Interestingly in ESCs, exons that are sensitive PTBP1 depletion are largely not responsive to the concomitant PTBP2 induction. This contrasts with NPCs where exons whose splicing shifts with PTBP1 depletion are often more strongly affected by PTBP1/2 co-depletion. We previously found that in mice heterozygous for the PTBP2 knockout mutation, certain exons were spliced at 50% the level seen in the homozygous null, indicating a strong effect of PTBP2 concentration. Similarly, we find here that certain exons, such as Gabbr1 exon 15, are very sensitive to moderate changes in PTBP1 expression, such as the reduction seen when ESCs differentiate into NPCs. Like PTBP2, the PTBP1 targets display a range of responsiveness to PTBP1 concentration, and the ESC system gives us a new tool for examining this earlier PTBP1 dependent program.

Alternative exons are usually regulated by ensembles of splicing factors acting to repress or activate their splicing (*Fu and Ares, 2014*; *Lee and Rio, 2015*). A question of interest is how targeting by the PTB proteins is affected by other neuronal splicing factors. Two known PTBP1 cofactors that may also affect PTBP2 are Matrin3 and Raver1 (*Coelho et al., 2015*; *Rideau et al., 2006*; *Huttelmaier, et al., 2001d*). Both these proteins are likely to affect the activity of PTBP1 and PTBP2 on certain targets. We find that both proteins are well expressed in ES cells, and Matrin3 is strongly upregulated with differentiation into NPCs and neurons. We recently identified several potential cofactors that alter the splicing of the PTBP1/2 target exon in Dlg4 (*Zheng et al., 2013*). These will also be interesting to examine in relation to additional PTBP targets, and whether they more strongly affect exons controlled by PTBP1, PTBP2 or both proteins. A protein that can counteract PTBP repression is nSR100/SRRM4, which is induced with neuronal differentiation and whose targets include some PTBP1/2 targets (*Calarco et al., 2009*; *Raj et al., 2014*). SRRM4 expression coincides with PTBP2, and its role may be to specifically antagonize the effects of PTBP2 on certain exons in immature neurons. It will be also interesting to identify the SRRM4 target exons during NPC differentiation and to assess their sensitivity to PTBP2 compared to PTBP1.

The intent of this study was to define a set of targets that are primarily responsive to PTBP1, and thus may affect early neuronal lineage commitment and differentiation. We identify a diverse group of transcripts that are sensitive to PTBP1 depletion from ESCs and which change their splicing when ESCs differentiate into NPCs and then into early neurons, as PTBP1 is reduced. Among these targets, we focused on Pbx1, which contains an alternative exon 7 that is highly responsive to PTBP1 concentration and is implicated in neuronal transcriptional regulation.

Using Cas9 genome editing, we created Pbx1 alleles that eliminate PTBP1 repression of exon 7, causing its premature splicing during development (*Doudna and Charpentier, 2014*; *Cong and Zhang, 2015e*). This early switch in Pbx1 isoforms was sufficient to induce a variety of transcripts, including multiple mRNAs implicated in neuronal lineage commitment. Thus, one function of PTBP1 is to repress the neuronal form of Pbx1 and prevent its action at target genes early in development. It will be interesting to further examine how different interactions of the two Pbx1 isoforms might lead to their different activities. Pbx1b lacks 83 amino acids at the C-terminus of Pbx1a but retains domains that facilitate interactions with the Hox, Prep, and Meis transcription factors (*Longobardi et al., 2013*). An earlier study found that Pbx1a and Pbx1b differ in their interactions with the transcriptional corepressors NCoR1 and NCoR2 and their ability to repress reporter gene expression (*Asahara et al., 1999b*). Our data indicates that Pbx1a may enhance Meis1 binding at the Hoxc5 locus to activate its expression. Thus, the changes in Pbx1 structure likely influence enhancer complex assembly at specific genes.

A splicing factor will often regulate a large network of targets that may overlap with the targets of other factors. The complexity of these regulatory programs is a challenge for understanding their biological roles. Mouse mutations in splicing regulators often exhibit lethal or highly pleiotropic phenotypes. Among the many affected transcripts, few isoforms are usually sufficiently analyzed to understand their altered activity. Cas9 genome editing offers an tool for examining differential isoform activity in the developing nervous system (*Gueroussov et al., 2015c*). Combining genome editing with stem cell differentiation allows the characterization of isoform activity in specific cells at

specific times in development, and provides means for investigating the function of other PTB targets within the larger programs of neurogenesis.

## Materials and methods

### Tissue culture

HB9-GFP and 46C mouse ESCs were gifts of H. Wichterle and T. Jessell, Columbia University, and of J. Sanford, UCSC, respectively. Cells were confirmed for expression of appropriate developmental and transgenic markers by immunostaining, and to be mycoplasma-free by PCR-based testing. 46C and HB9-GFP mouse ESCs were grown on 0.1% gelatin-coated dish with CF1 mouse embryonic fibroblasts (Applied StemCell, Inc., Menlo Park, CA) in ESC media. ESC media consisted of DMEM (Fisher Scientific, Hampton, NH) supplemented with 15% ESC-qualified fetal bovine serum (Life Technologies, Carlsbad, CA), 1X non-essential amino acids (Life Technologies), 1X GlutaMAX (Life Technologies), 1X ESC-qualified nucleosides (EMD Millipore, Billerica, MA), 0.1 mM 2-Mercaptoethanol (Sigma-Aldrich, St. Louis, MO), and 1000U/ml ESGRO leukemia inhibitor factor (EMD Millipore).

46C and HB9-GFP ESCs were differentiated into NPCs according to (Conti et al., 2005), with minor modifications. Briefly, feeder-free ESCs were differentiated on 0.1% gelatin-coated dish in N2B27 media. N2B27 media consisted of 1:1 mixture of Neurobasal media (Life Technologies) and DMEM/F12 (Fisher Scientific) supplemented with 0.5X B27 without Vitamin A (Life Technologies), 0.5X N2 Supplement (Life Technologies), 1X GlutaMAX (Life Technologies), and 0.1 mM 2-Mercaptoethanol (Sigma-Aldrich). After seven days of differentiation, cells were trypsinized and grown in aggregate culture for two days in N2B27 supplemented with 10 ng/ml recombinant human EGF (PeproTech, Rocky Hill, NJ) and 10 ng/ml recombinant human FGF-basic (PeproTech). Aggregates were plated on poly-ornithine-coated (15 ug/ml, Sigma) and fibronectin-coated (1.5 ug/ml, Sigma-Aldrich) dishes in NPC media. NPC media consisted of DMEM/F12 (Fisher Scientific) supplemented 1X B27 without Vitamin A (Life Technologies), 1X GlutaMAX (Life Technologies), 0.1 mM 2-Mercaptoethanol (Sigma-Aldrich), 10 ng/ml recombinant human EGF (PeproTech), and 10 ng/ml recombinant human FGF-basic (PeproTech). NPCs were maintained on poly-ornithine-coated (15 ug/ml, Sigma-Aldrich) and fibronectin-coated (1.5 ug/ml, Sigma-Aldrich) dishes in NPC media for one to two passages. NPCs were differentiated with the removal of EGF and FGF, along with the addition of 1 uM all-trans retinoic acid (RA, Sigma-Aldrich).

HB9-GFP ESCs were differentiated to MNs according to (Wichterle et al., 2002), with minor modifications. Briefly, feeder-free ESCs were grown as aggregate culture for two days in MN media. MN media consisted of 1:1 mixture of Neurobasal media (Life Technologies) and DMEM/F12 (Fisher Scientific) supplemented with 10% Knockout Serum Replacement (Life Technologies), 1X GlutaMAX (Life Technologies), and 0.1 mM 2-Mercaptoethanol (Sigma-Aldrich). After two days, the MN media was supplemented with 1X N2 Supplement (Life Technologies), 1 uM all-trans retinoic acid (RA, Sigma-Aldrich), and 1 uM smoothened agonist (EMD Millipore). After five days of RA and SAG addition, EBs were plated on BD Matrigel (BD Biosciences), with the addition of 10 ng/ml BDNF (R&D Systems, Minneapolis, MN), 10 ng/ml GDNF (R&D Systems), and 10 ng/ml CNTF (R&D Systems).

### Cell sorting of GFP+ MNs

EBs were collected 5 or 8 days after RA and SAG addition. EBs were dissociated in Acutase (Innovative Cell Technologies, San Diego, CA) for 10 min and sorted using a BD FACSAria cell sorter at the UCLA Broad Stem Cell center core facility.

### siRNA transfections

46C ESCs and NPCs were transfected with Silencer Select Ptbp1 (Life Technologies; s72335, s72336) and Ptbp2 (Life Technologies; s80149, s80149) siRNAs using RNAiMax (Life Technologies) according to the manufacturer's recommendations. Silencer Negative Control siRNA #1 (Life Technologies) and siGENOME Non-Targeting siRNA Pool #1 (GE Healthcare, Pittsburg, PA) were used as negative controls. ESCs were reverse transfected with 20 nM of siRNAs, treated again 24 hr later, and collected 72 hr post-transfections. NPCs were reverse transfected with 10 nM of siRNAs and collected 48 hr post-transfections.

## Western blots and immunocytochemistry

Western blots were performed on total protein from cell cultures lysed in RIPA buffer supplemented with protease inhibitors (Roche, Basel, Switzerland) and Benzonase (Sigma-Aldrich). Lysates were diluted in 4X SDS loading buffer, heated for at 95°C for 10 min, and loaded onto 10% polyacrylamide Laemmli SDS PAGE gels. Gels were run under standard electrophoresis conditions. Transfers were performed on a Novex X-Cell mini-cell transfer apparatus (Life Technologies) onto Immobilon-FL PVDF membranes (EMD Millipore). The membranes were probed under standard conditions with primary antibodies overnight at 4°C. The following primary antibodies were used: rabbit anti-PTBP1 antibody PTB-NT (1:3000) (*Markovtsov et al., 2000*), rabbit anti-PTBP2 antibody nPTB-IS2 (1:1000) (*Sharma et al., 2005*), rabbit anti-Pbx1 antibody (1:500, Cell Signaling, Danvers, MA), rabbit anti-U170K antibody (1:1000) (*Sharma et al., 2005*), and mouse anti-GAPDH 65C antibody (1:10000, Life Technologies). For fluorescent detection, the membranes were probed with ECL Plex Cy3 and Cy5-conjugated goat anti-mouse and goat anti-rabbit secondary antibodies (1:2000; GE Healthcare). The blots were then scanned on a Typhoon Phosphorimager (GE Healthcare) and quantified using ImageQuant software (GE Healthcare). For chemiluminescent detection, the membranes were probed with Amersham ECL HRP Conjugated Antibodies (1:4000, GE Healthcare) and developed using SuperSignal West Femto Maximum Sensitivity Substrate (Life Technologies) and Kodak BioMax XAR film (Sigma-Aldrich).

Cell cultures were fixed in 4% paraformaldehyde (Electron Microscopy Sciences, Hatfield, PA). Adherent fixed cells were incubated in permeabilization buffer (PBS, 0.25% Triton X-100), washed with block solution (PBS, 0.1% Triton X-100, 1% bovine serum albumin), and probed with primary antibodies overnight at 4°C. Fixed EB cell cultures were cryoprotected in 30% sucrose-PBS, frozen in Tissue-Tek OCT (Electron Microscopy Sciences), and stored at −80°C until use. 10 µM sections were prepared on a cryostat. Sections were washed with block solution, and probed with primary antibodies overnight at 4°C. The following primary antibodies were used: chicken anti-GFP antibody (1:1000, Abcam, Cambridge, United Kingdom), rabbit anti-HB9 antibody (1:8000) (*Arber et al., 1999c*), rabbit anti-PTBP2 antibody nPTB-IS2 (1:1000), mouse anti-Nestin antibody (1:250, Developmental Studies Hybridoma Bank, Iowa City, IA), and mouse-anti TuJ1 antibody (1:1000, Covance, Princeton, NJ). Samples were washed in PBS before incubation with Alexa-conjugated secondary antibodies (Life Technologies) for 2 hr at room temperature. Samples were rinsed in PBS and mounted with Prolong Gold AntiFade containing nuclear stain DAPI (Life Technologies). Images were acquired using the Carl Zeiss Laser Scanning System LSM 510 META confocal microscope.

## RNA isolation, RT-PCR, and real-time PCR

Total RNA was collected from cell cultures using Trizol (Life Technologies) according to the manufacturer's instructions. RNA was quantified (A260) using a Nanodrop-1000 spectrophotometer (Nanodrop Technologies/Thermo Fisher, San Jose, CA). Total RNA (0.5–1 µg) was used for each sample in a 10 µL reaction with 0.25 µL of SuperScript III RT (Life Technologies). PCR reactions contained 200,000–500,000 counts per minute of 32P-labeled or 0.2 µM FAM-labeled reverse primer. PCR reactions were run for 24 cycles for ESCs, 19 cycles for NPCs, and 20 cycles for MN cultures at an annealing temperature of 60°C. PCR reactions were mixed 1:2 with 95% formamide and loaded onto 8% polyacrylamide, 7.5 M urea gels. Gels were run under standard electrophoresis conditions, dried, and imaged on a Typhoon Imager (GE Healthcare). Bands were quantified with ImageQuant software (GE Healthcare). Real-time PCR was performed using SensiFAST SYBER Lo-ROX Kit (Bioline, London, United Kingdom) on a QuantStudio 6 Real-Time PCR System (Life Technologies) according to the manufacturers' instructions. Relative mRNA levels were determined using a standard curve with beta-actin as a normalizing control. A list of primer sequences is available in *Supplementary file 1*.

## Generation of CRISPR cell lines

Guide RNAs targeting Pbx1 intron 6 were designed using http://crispr.mit.edu/ and cloned into a modified pX330-U6-Chimeric_BB-CBh-hSpCas9 plasmid (gift of B. Stahl and J. Doudna, University of California, Berkley) according to the published protocol (*Ran et al., 2013*). To generate Cas9-targeted deletions, HB9-GFP ESCs were transfected with two modified pX330 constructs and pBABE-puro (Addgene, Cambridge, MA; #42230) using BioT (Bioland Scientific, Paramount, CA) according

to the manufacturer's recommendations. Transfected cells were then treated with 0.5 ug/ml puromycin (InVivoGen, San Diego, CA). Following puromycin selection, clonal ESC lines were isolated and genotyped for intron 6 deletions. Some intron 6 deletions using guides 1 and 4 resulted in the loss of splice sites needed for exon 7 splicing. After genotyping to identify intron 6 deletions, ESC clones were subjected to secondary tests to confirm exon 7 splicing. A list of guide RNA and genotyping primer sequences is in *Supplementary file 1*.

## iCLIP sequencing and data analysis

46C ESCs and NPCs were cross-linked at 100 mJ/cm2. Cell pellets were collected and flash-frozen for iCLIP library preparation (*König et al., 2010*). Modifications to the protocol are described in *Supplementary file 2*. Flag and PTBP1 libraries were subjected to 100 bp single-end sequencing at the UCLA Broad Stem Cell center core facility (Illumina HiSeq2000). Data analyses were performed according to (*König et al., 2010*), with minor modifications. Briefly, PCR duplicates were removed by comparing random portions of the sequenced barcodes. Unique reads were mapped to the mouse genome (mm9/NCBI37) using Bowtie, allowing up to two nucleotide mismatches (*Langmead et al., 2009c*). Mapped reads were further mapped to the longest transcripts in Known Gene table (*Hsu et al., 2006d*). Crosslink sites were defined as the nucleotide upstream of each iCLIP read, and the False Discovery Rate (FDR) calculated according to (*König et al., 2010*), with significant crosslink sites (FDR<0.01) used for clustering and downstream analyses. Significant crosslinking sites were extended 20 nucleotides on either side, and overlapping sites compiled into clusters. Genomic sequences 30 nucleotides upstream and downstream from each crosslink site were used for motif enrichment analyses. Z-scores for all pentamer motifs were calculated by comparing motif frequencies relative to randomly chosen intervals from the same introns. Cassette exons were evaluated for PTBP1 clusters containing $\geq$ four significant reads, and defined as likely targets if a cluster was present within the cassette exon or within the intron sequence 500 nt upstream or downstream.

## RNA sequencing and data analysis

To identify splicing changes during neuronal differentiation, polyA-plus RNA was isolated from HB9-GFP ESCs, NPCs, and Day 5 GFP+ MNs. Paired-end libraries were constructed using the TruSeq mRNA Library Prep Kit (Illumina, San Diego, CA) with modifications to generate strand-specific libraries (*Li et al., 2014*). The libraries were subjected to 100 nt paired-end sequencing at the UCLA Broad Stem Cell center core facility (Illumina HiSeq2000). Data were analyzed using the Cufflinks pipeline to generate 200–250 million mapped reads per sample (*Trapnell et al., 2012*). Alternative exon inclusion levels were determined by mapping to exon duo and trio databases using SpliceTrap (*Wu et al., 2011*). Gene Ontology analyses were performed using DAVID (*Huang et al., 2009a*).

To identify splicing changes following PTB protein depletion, polyA-plus RNA was isolated from 46C ESCs treated with siControl, siPtbp1, or siPtbp1/2 (n=3); and 46C NPCs treated with siControl, siPtbp1, siPtbp1/2, or siPtbp2 (n=2). Paired-end, strand-specific libraries were constructed using the TruSeq Stranded mRNA Library Prep Kit (Illumina). The libraries were subjected to 50 nt paired-end sequencing (Illumina HiSeq2000) to generate 30 to 50 million mapped reads per replicate. The replicates were then pooled for SpliceTrap analyses (*Wu et al., 2011*).

To identify gene expression changes with Pbx1a induction, polyA-plus RNA was isolated from HB9-GFP Day 2 MNs differentiated from wild type (n=3) and I6 +/- (n=5) ESC clones. Sequencing libraries were constructed using the TruSeq Stranded mRNA Library Prep Kit (Illumina) and subjected to 50 nt single-end sequencing (Illumina HiSeq2000) to generate 25 to 40 million mapped reads per replicate. Differential gene expression was calculated using the Cufflinks pipeline (q-value < 0.05). Normalized FPKM values were calculated by Cuffnorm and were used to perform a Welch's t-test (P-value <0.05). Gene Ontology analyses were performed using DAVID (*Huang et al., 2009a*). Differentially expressed genes were evaluated for Pbx1 ChIP clusters (*Penkov et al., 2013b*). Pbx1 binding was determined if a ChIP cluster was present within a window 1 kb upstream and downstream of the gene locus.

## ChIP-qPCR

HB9-GFP EBs were collected 2 days after RA and SAG addition, and dissociated in 0.25% Trypsin (Life Technologies) for 10 min. Cell pellets were collected and flash-frozen for ChIP-qPCR. Cells were

cross-linked using 1% methanol-free formaldehyde, and chromatin was isolated using a previously published protocol (*Schjerven et al., 2013c*). ChIP was performed using antibodies against Pbx1 (Cell Signaling), Meis1 (AbCam), and Prep1/Pknox1 (Thermo Fisher). Primers for qPCR are provided in *Supplementary file 3*.

Datasets are submitted to GEO with Accession Number GSE71179, downloadable at: http://www.ncbi.nlm.nih.gov/geo/query/acc.cgi?token=stmpwqeyppetvkt&acc=GSE71179

A list of Genome Browser sessions is provided in *Supplementary file 4.*

## Acknowledgements

We thank Hynek Wichterle and Tom Jessell for the HB9-GFP cell line, Xin Liu for help with the ChIP-qPCR assays, and Brett Staahl and Jennifer Doudna for the CRISPR-Cas9 plasmid. B.G.N. was supported by the UCLA Broad Center for Regenerative Medicine and Stem Cell Research (BSCRC), The Rose Hills Foundation, and grants from the NINDS (NS072804) and the California Institute for Regenerative Medicine (CIRM, RB1-01367). K.L.A. was supported by the UCLA Cellular and Molecular Biology Training program (Ruth L. Kirschstein NIH GM00785), the UCLA-California Institute for Regenerative Medicine Training Grant and a UCLA Graduate Division Dissertation Year Fellowship. Work in the D.L.B lab was supported by NIGMS (R01 GM49662), CIRM (RB2-01502), the BSCRC, and the Howard Hughes Medical Institute. AJL was supported by the UCLA MSTP program, and the training programs in Neural Repair (T32 NS07449) and Neurobehavioral Genetics (T32 NS048004) at UCLA.

## Additional information

### Competing interests

DLB: Reviewing editor, *eLife*. The other authors declare that no competing interests exist.

### Funding

| Funder | Grant reference number | Author |
| --- | --- | --- |
| Howard Hughes Medical Institute | | Douglas L Black |
| National Institute of General Medical Sciences | R01 GM49662 | Douglas L Black |
| National Institute of Neurological Disorders and Stroke | NS072804 | Bennett G Novitch |
| California Institute for Regenerative Medicine | RB1-01367; RB2-01502 | Bennett G Novitch Douglas L Black |
| UCLA Broad Center for Regenerative Medicine and Stem Cell Research | | Bennett G Novitch Douglas L Black |
| National Institute of General Medical Sciences | GM00785 | Katrina L Adams |
| National Institute of Neurological Disorders and Stroke | NS048004 | Anthony J Linares |
| National Institute of Neurological Disorders and Stroke | NS07449 | Anthony J Linares |

The funders had no role in study design, data collection and interpretation, or the decision to submit the work for publication.

### Author contributions

AJL, Conception and design, Acquisition of data, Analysis and interpretation of data, Drafting or revising the article; C-HL, Analysis and interpretation of data, Drafting or revising the article; AD, Acquisition of data, Analysis and interpretation of data, Drafting or revising the article, Contributed

unpublished essential data or reagents; KLA, Acquisition of data, Drafting or revising the article, Contributed unpublished essential data or reagents; BGN, Conception and design, Analysis and interpretation of data, Drafting or revising the article, Contributed unpublished essential data or reagents; DLB, Conception and design, Analysis and interpretation of data, Drafting or revising the article

### Author ORCIDs
Anthony J Linares, http://orcid.org/0000-0002-2484-7252
Douglas L Black, http://orcid.org/0000-0002-2705-8187

## Additional files

### Supplementary files
• Supplementary File 1. Primer and guide RNA sequences.

• Supplementary File 2. iCLIP protocol.

• Supplementary File 3. Primer sequences for ChIP-qPCR.

• Supplementary File 4. RNA-seq and iCLIP genome browser sessions.

### Major datasets
The following dataset was generated:

| Author(s) | Year | Dataset title | Dataset URL | Database, license, and accessibility information |
|---|---|---|---|---|
| Lin c | 2016 | The splicing regulator PTBP1 controls the activity of the transcription factor Pbx1 during neuronal differentiation | http://www.ncbi.nlm.nih.gov/geo/query/acc.cgi?acc=GSE71179 | Publicly available at the NCBI Gene Expression Omnibus (Accession no: GSE71179). |

The following previously published dataset was used:

| Author(s) | Year | Dataset title | Dataset URL | Database, license, and accessibility information |
|---|---|---|---|---|
| Penkov D, Mateos San Martín D, Fernandez-Díaz LC, Rosselló CA, Torroja C, Sánchez-Cabo F, Warnatz HJ, Sultan M, Yaspo ML, Gabrieli A, Tkachuk V, Brendolan A, Blasi F, Torres M | 2013 | TALE transcription factors binding and transcriptional effect | http://www.ncbi.nlm.nih.gov/geo/query/acc.cgi?acc=GSE39609 | Publicly available at the NCBI Gene Expression Omnibus (Accession no: GSE39609). |

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
