## [Decision Letter]

Thank you for submitting your work entitled "The splicing regulator PTBP1 controls the activity of the transcription factor Pbx1 during neuronal differentiation" for peer review at *eLife*. Your submission has been overall favorably evaluated by James Manley (Senior Editor) and three reviewers, one of whom is a member of our Board of Reviewing Editors.

The reviewers have discussed the reviews with one another and the Reviewing editor has drafted this decision to help you prepare a revised submission.

Summary:

In this manuscript Linares et al. investigate the contributions of the splicing regulators PTBP1 and PTBP2 to alternative splicing programs during differentiation of embryonic stem cells (ESCs) into motor neurons (MNs). They characterize the expression dynamics and contributions of these factors to splicing regulation during different stages of differentiation. They then investigate the function of a conserved PTBP1/2-regulated target, exon 7 of the homeodomain transcription factor Pbx1. Derepression by PTBP1 results in increased inclusion of exon 7 and expression of the Pbx1a isoform during differentiation. By utilizing the CRISPR/Cas9 system, the authors delete a PTBP1 intronic splicing repressor element proximal to exon 7. Using RNA-Seq analysis, the authors show that this results in earlier activation of the Pbx1a isoform, and activation of Pbx1 target genes, during MN differentiation.

Overall, this is an interesting manuscript that provides detailed insight into the splicing programs regulated by PTBP1/2 during MN development, as well as information on the function of a specific exon target in this program that is thought to affect the activity of Pbx1. Alternative splicing regulators typically control large numbers of target exons and introns during development, yet the functions of the vast majority of these targets events are not known. Characterizing the contributions of individual regulated splicing events to differentiation programs is therefore of considerable interest. The reviewers have the following recommendations for revisions.

Major:

Much of the first half of the manuscript (i.e., Figure 1–Figure 3) presents results that essentially recapitulate those already published by Dr. Black and his colleagues (and by others) in other neural developmental systems. In our view, the manuscript would benefit from condensing the presentation of this section and further developing the investigation of the function of Pbx1 isoforms. In this regard, the authors are requested to address the following:

a) The RNA-Seq data generated for the splicing analysis can also be used to look at steady-state mRNA levels during the different stages of differentiation. Do the genes induced by Pbx1a expression listed in Figure 6 also show increased expression?

b) The authors cite previous ChIP-Seq data indicating that some of the genes affected by modulation Pbx1 exon 7 are direct Pbx1 targets, but these experiments were performed in a different context. It would strengthen the manuscript to show that Pbx1 binds directly to these targets during MN differentiation. This could be achieved by performing ChIP-qPCR assays, although ChIP-Seq would be preferable. Moreover by performing a ChIP analysis at Day 2 in their CRISPR-Cas9 edited lines the authors could assess differences in DNA binding between the isoforms.

c) Related to (b), it would be informative to assess whether the two Pbx1 isofoms differ in their ability to bind partner proteins. Time permitting, the authors could perform ChIP-qPCR assays on known Pbx1 co-factors at target genes to test whether the isoforms differ in their ability to recruit known binding partners.

---

## [Author Response]

*[…] Much of the first half of the manuscript (i.e., Figure 1–Figure 3) presents results that essentially recapitulate those already published by Dr. Black and his colleagues (and by others) in other neural developmental systems. In our view, the manuscript would benefit from condensing the presentation of this section and further developing the investigation of the function of Pbx1 isoforms.*

We respectfully disagree that these analyses simply recapitulate previous work. It is of course well known from work of multiple labs that loss of PTBP1 induces PTBP2 expression during neuronal development. The program of splicing controlled by PTBP2 has been characterized in the mouse by ourselves and Licatalosi et al. However, the program controlled by PTBP1 has been difficult to define in mice due to embryonic lethality, and the other systems where PTBP1 has specifically been examined are not of clear relevance to actual neuronal development. There are studies of PTBP dependent exons in cell lines such as Hela, usually defined as responsive to double PTBP1/PTBP2 knockdown. There are the very interesting results from the Fu lab showing transdifferentiation of MEF and other cell lines into neuronal-like cells in response to PTBP1 knockdown. It is difficult to relate these results to normal neuronal development because MEFs do not express the same gene set as true pluripotent progenitor cells, and it is not clear whether their transdifferentiation goes through an NPC-like intermediate. The intent of our study was to define the targets of PTBP1 regulation within a well-characterized pathway of differentiation into actual neuronal progenitor cells and then into a defined neuronal cell type, motor neurons. Targets defined in this system can then be assessed relative to neuronal development in the animal. Indeed, we find that ESC do not simply differentiate into neurons upon loss of PTBP1, as was seen in MEFs. However, neuronal gene expression programs are clearly upregulated under these conditions. Since our last submission, Benjamin Blencowe has also published data examining PTBP1 expression in differentiating ESC. This paper (Gueroussov et al. 2015) shows a very nice profile of PTBP1 expression across differentiation, and then goes on to focus on activity differences between PTBP1 isoforms during neuronal differentiation rather than differences between PTBP1 and PTBP2. By comparing ESC cells to NPC, we have identified a set of exons that are more responsive to PTBP1 than PTBP2. These were difficult to identify in cell lines and in the mouse, and using these data we can now begin to examine how the PTBP1 and PTBP2 regulatory programs might differ. These exons are also a resource for those of us interested in mechanistic questions of how differential targeting by related splicing factors is achieved. Thus, these results provide an essential foundation not available from the prior work, both for the subsequent experiments in the paper that begin to analyze the PTBP1 program, and for many future studies.

We have rewritten the Abstract and added text in multiple places to make clearer how these studies address new questions regarding PTBP regulation.

*In this regard, the authors are requested to address the following: a) The RNA-Seq data generated for the splicing analysis can also be used to look at steady-state mRNA levels during the different stages of differentiation. Do the genes induced by Pbx1a expression listed in Figure 6 also show increased expression?*

We should have discussed this before. The CuffDiff tables of expression as ESCs differentiate into NPCs and GFP+ MNs are now presented as [Supplementary-material SD10-data]. As we now discuss in subsection “The early expression of Pbx1a promotes neuronal gene expression”, the genes most strongly induced by shifting splicing to Pbx1a are also upregulated during differentiation of ESC to NPC and MNs. These include many genes with well established roles in neuronal development such as Hoxc5.

*b) The authors cite previous ChIP-Seq data indicating that some of the genes affected by modulation Pbx1 exon 7 are direct Pbx1 targets, but these experiments were performed in a different context. It would strengthen the manuscript to show that Pbx1 binds directly to these targets during MN differentiation. This could be achieved by performing ChIP-qPCR assays, although ChIP-Seq would be preferable. Moreover by performing a ChIP analysis at Day 2 in their CRISPR-Cas9 edited lines the authors could assess differences in DNA binding between the isoforms. c) Related to (b), it would be informative to assess whether the two Pbx1 isofoms differ in their ability to bind partner proteins. Time permitting, the authors could perform ChIP-qPCR assays on known Pbx1 co-factors at target genes to test whether the isoforms differ in their ability to recruit known binding partners.*

We performed a series of ChIP-qPCR experiments to examine Pbx1 and cofactor binding. As suggested, these were performed at Day 2 of differentiation in the WT and I6+/- cells. Briefly, we demonstrate binding of Pbx1 on 6 target genes whose expression goes up with Pbx1a. We do not observe differences in DNA binding by Pbx1 between the mutant and wild type cells. This may not be surprising given that the DNA binding domain of the protein is not altered by the splicing change. We went on to perform ChIP-qPCR of the Meis1 and Prep1 co-factors for Pbx1. These are also seen to bind the target genes. In one case we see an increase in binding of the Meis1 cofactor on the target gene Hoxc5, when expression is shifted to the Pbx1a isoform. These data are now presented in Figure 6 and Figure 6—figure supplement 1, and discussed in the Results and Discussion sections.